# Temperature dependence of spherical electron transfer in a nanosized [Fe$_{14}$] complex

Wei Huang[1], Shuqi Wu[2], Xiangwei Gu[1], Yao Li[1], Atsushi Okazawa [3], Norimichi Kojima[4], Shinya Hayami [5], Michael L. Baker [6,7], Peter Bencok[8], Mariko Noguchi[9,12], Yuji Miyazaki[9], Motohiro Nakano [9], Takumi Nakanishi[2], Shinji Kanegawa[2], Yuji Inagaki [10], Tatsuya Kawae[10], Gui-Lin Zhuang [11], Yoshihito Shiota [2], Kazunari Yoshizawa[2], Dayu Wu[1]* & Osamu Sato[2]*

The study of transition metal clusters exhibiting fast electron hopping or delocalization remains challenging, because intermetallic communications mediated through bridging ligands are normally weak. Herein, we report the synthesis of a nanosized complex, [Fe(Tp) (CN)$_3$]$_8$[Fe(H$_2$O)(DMSO)]$_6$ (abbreviated as [Fe$_{14}$], Tp$^-$, hydrotris(pyrazolyl)borate; DMSO, dimethyl sulfoxide), which has a fluctuating valence due to two mobile $d$-electrons in its atomic layer shell. The rate of electron transfer of [Fe$_{14}$] complex demonstrates the Arrhenius-type temperature dependence in the nanosized spheric surface, wherein high-spin centers are ferromagnetically coupled, producing an $S = 14$ ground state. The electron-hopping rate at room temperature is faster than the time scale of Mössbauer measurements (<~10$^{-8}$ s). Partial reduction of N-terminal high spin Fe$^{III}$ sites and electron mediation ability of CN ligands lead to the observation of both an extensive electron transfer and magnetic coupling properties in a precisely atomic layered shell structure of a nanosized [Fe$_{14}$] complex.

[1] Jiangsu Key Laboratory of Advanced Catalytic Materials and Technology, Advanced Catalysis & Green Manufacturing Collaborative Innovation Center, School of Petrochemical Engineering, Changzhou University, Changzhou 213164, China. [2] Institute for Materials Chemistry and Engineering & IRCCS, Kyushu University, 744 Motooka, Nishi-ku, Fukuoka 819-0395, Japan. [3] Department of Basic Science, Graduation School of Arts and Sciences, The University of Tokyo, 3-8-1 Komaba, Meguro-ku, Tokyo 153-8902, Japan. [4] Toyota Physical and Chemical Research Institute, Yokomichi, Nagakute, Aichi 480-1192, Japan. [5] Department of Chemistry, Graduate School of Science and Technology and Institute of Pulsed Power Science (IPPS), Kumamoto University, 2-39-1 Kurokami, Chuo-ku, Kumamoto 860-8555, Japan. [6] The School of Chemistry, The University of Manchester, Manchester M13 9PL, UK. [7] The School of Chemistry, The University of Manchester at Harwell, Didcot OX11 0FA, UK. [8] Diamond Light Source, Science Division, Didcot OX11 0DE, UK. [9] Research Center for Structural Thermodynamics, Graduate School of Science, Osaka University, Toyonaka, Osaka 560-0043, Japan. [10] Department of Applied Quantum Physics, Faculty of Engineering, Kyushu University, 744 Motooka, Nishi-ku, Fukuoka 819-0395, Japan. [11] Institute of Industrial Catalysis, College of Chemical Engineering, State Key Lab Breeding Base of Green-Chemical Synthesis Technology, Zhejiang University of Technology, Hangzhou 310032, China. [12]Present address: Department of Chemistry, College of Humanities and Sciences, Nihon University, 3-25-40 Sakurajosui, Setagaya-ku, Tokyo 156-8550, Japan. *email: wudy@cczu.edu.cn; sato@cm.kyushu-u.ac.jp

To date, numerous nanosized transition metal complexes have been synthesized[1], which can be predetermined to have various fascinating structures via molecular self-assembly process[2,3]. One of the challenges in realizing their functions is the introduction of fast electron transfer over a long distance that ubiquitously occurs in chemical and biological systems[4,5]. The low-nuclearity compounds have been intensively investigated as intramolecular electron transfer systems[6–13]. The recent interesting work on the $[Fe_2]^V$ mixed-valence compound exhibited changes in the electron transfer rate observable by Mössbauer spectroscopy with a ferromagnetic $S = 9/2$ ground state[14]. It is still challenging to isolate nanoarchitectures where electron transfer occurs over more metal centers, toward revealing extraordinary spectral and electronic behavior[15]. In the presence of intermetallic coupling in high-nuclearity complexes, the number of hopping $d$ electrons and their interactions would be expected to increase. This would be accompanied by some intriguing processes, such as electron transfer along topology-specific pathways, many-electron transfer processes, and externally induced charge separation in a confined nanospace[16–18]. However, intermetallic electron transfer-mediated through bridging ligands are normally weak; thus, properties related to confined electron transfer processes in discrete nanosized complexes have remained hypothetical thus far[19,20].

Prussian blue analogs have received considerable attention as they can demonstrate the coexistence of electron transfer and exchange interaction through a cyanide bridge[21,22]. The realization of partially reducing or oxidizing the interacting metal centers in a nanoarchitecture is proposed as a route to achieve the extensive multielectron transfer with novel magnetic and electronic properties[23,24]. Here, we report a cyanide-bridged complex $[Fe^{II}(Tp)(CN)_3]_8[Fe^{2.667+}(H_2O)(DMSO)]_6$ (abbreviated as $[Fe_{14}]$, $Tp^- =$ hydrotris(pyrazolyl)borate; DMSO = dimethyl sulfoxide), in which two extra $3d$ electrons hop at a rate faster than the time scale of Mössbauer measurements ($<\sim 10^{-8}$ s) at room temperature. The key feature of this complex is that only two of the six N-terminal Fe-hs sites in the $Fe^{II–ls}$–CN–$Fe^{hs}$ structure are successfully reduced (hs = high spin and ls = low spin), thereby reducing the potential barrier for two-electron hopping at the six N-terminal Fe-hs sites. Furthermore, the rates of the intramolecular electron transfer exhibit a distinct temperature dependence that can be described with an Arrhenius law in the nanosized complex. Another important characteristic is that a ferromagnetic interaction operates in $[Fe_{14}]$ with a ground spin $S = 14$, which implies that extra $3d$ electrons hop around the exchange-coupled atomic layer shell.

## Results

**Crystal structure**. The synthesis of the $[Fe_{14}]$ complex proceeds via the reaction of $Fe(BF_4)_2 \cdot 6H_2O$ with $Bu_4N[Fe(Tp)(CN)_3]$ in a mixture of water and DMSO. The crystalline sample of $[Fe_{14}]$ is thermally stable at room temperature (Supplementary Figs. 1 and 2). The crystal structure of $[Fe_{14}]$ features a tetradecanuclear iron molecular cluster, $[Fe(Tp)(CN)_3]_8[Fe(H_2O)(DMSO)]_6$, as shown in Fig. 1a, b (Supplementary Tables 1 and 2), wherein Fe site in $[Fe(Tp)(CN)_3]$ and $[Fe(H_2O)(DMSO)]$ is hereafter referred to as the A and B site, respectively (Fig. 1c, d). Each $[Fe(Tp)(CN)_3]$ unit at A site is connected to three B-site Fe ions via a facial triad of cyanide-bridged units, where the B-site Fe center is in a weakly distorted octahedron coordinated by axial water and DMSO molecules. The equatorial coordination of B-site Fe is completed by the four N-terminals of cyanide ligands with an average Fe···N bond distance of 2.068(6) Å. Thus, eight $[Fe(Tp)(CN)_3]$ building units are symmetrically located at the vertices (A site) of a cubic cage, and six $[Fe(DMSO)(H_2O)]$ units occupy six centroid areas

(B site), resulting in the face-capped cubic structure. When we consider only the mode of the Fe–C≡N–Fe linkage in the framework, the simplified $[Fe_{14}]$ cage comprises 62 atoms $(Fe_{14}C_{24}N_{24})$ with a shell having an atomic layer thickness. The diameter is ca. 8.8 Å, which is slightly larger than that of $C_{60}$ (ca. 7.1 Å)[25].

**Electronic behavior**. To confirm the valences and spin states of the Fe ions in the complex, $^{57}Fe$ Mössbauer spectroscopic measurements were performed at various temperatures under zero magnetic field (Fig. 2a). A $^{57}Fe$ isotopically enriched sample, $[Fe_8{}^{57}Fe_6]$, was also prepared, consisting of isotopically enriched B sites ($^{57}Fe$, 96%) and naturally abundant A sites ($^{57}Fe$, 2%), to mainly produce Mössbauer signals of almost only the B sites. (Fig. 2b) Compared with the enriched sample, the natural $[Fe_{14}]$ sample exhibits one additional quadrupole doublet in the spectrum at both high and low temperatures. Considering its isomer shift (IS) of 0.07–0.15 mm s$^{-1}$ and quadrupole splitting (QS) of ~0.45 mm s$^{-1}$, the additional doublet can be assigned to $Fe^{II}$-ls in the $[Fe(Tp)(CN)_3]$ unit[21]. The result indicates that the reactant, $[Fe^{III}(Tp)(CN)_3]^-$, is completely reduced to $[Fe^{II}(Tp)(CN)_3]^{2-}$ during the formation of $[Fe_{14}]$. Furthermore, a comparison of the Mössbauer spectra of $[Fe_{14}]$ and $[Fe_8{}^{57}Fe_6]$ indicates that the role of the $Fe^{II}$ ions at the A sites remains static ($Fe^{II}$-ls) across the entire temperature range. (Supplementary Tables 3 and 4) In the spectra of the site-selective $^{57}Fe$ isotopically enriched sample, $[Fe_8{}^{57}Fe_6]$, exclusive information can be observed for the B sites, indicating that there is no electron transfer between the A and B sites in $[Fe_{14}]$. The spectra recorded below 144 K can be split into two doublets with QS values of 2.20 and 1.18 mm s$^{-1}$ (at 10.5 K) with a 1:2 ratio of their fractions, assignable to $Fe^{II}$-hs and $Fe^{III}$-hs, respectively. This finding demonstrates the apparent mixed-valence nature of the B site with the expected fraction of 2:4 for $Fe^{II}$ vs. $Fe^{III}$. Above 164 K, the B-site signals show a pronounced temperature dependence, especially at around 200 K. The signals merge gradually into a unique doublet upon an increase in the temperature, indicating that valence fluctuation occurs due to electron hopping at the B sites. When two extra electrons are shared with the six B-site Fe ions, the averaged B-site Fe valence is expected to be 2.667 for $[Fe^{II}_8]^A[Fe^{III}_6 + 2e]^B$. To evaluate such a fluctuation, an electron-hopping relaxation model was applied to the merged spectra between 164 and 297 K for $[Fe_8{}^{57}Fe_6]$[26]. As a result, the well-fitted profiles were produced with the $Fe^{II}$ vs. $Fe^{III}$ area ratio fixed at 2:4. A few additional parameter constraints were necessary to prevent divergence in the calculations for the spectra at 245−297 K because of strong parameter correlations, especially between the linewidth and relaxation rate. The results indicate that the electron-hopping rate at room temperature is comparable to or faster than the fast-exchange limit (~10$^{-8}$ s) of the Mössbauer time window. On the other hand, the fluctuation rate below 144 K is slower than the lower limit ($\tau = \sim 3 \times 10^{-7}$ s) of the time window, and the electron-hopping relaxation model no longer describes the system (Fig. 2c).

Figure 2d summarizes the QS and IS values from the spectral analysis. A sudden change in the temperature dependences of IS and QS for $Fe^{II}$-hs and $Fe^{III}$-hs was observed at around 220 K, whereas there was no such abrupt change for $Fe^{II}$-ls. This change seems to correlate with the electron hopping at the B sites of the complex. On cooling, the IS values of $Fe^{III}$-hs and $Fe^{II}$-ls increased almost linearly according to a second-order Doppler shift, except for the sudden change. In contrast, the IS values of $Fe^{II}$-hs anomalously rise upon an increase in the temperature and do not obey a second-order Doppler shift effect at high temperatures. However, the average $Fe^{II}$-hs and $Fe^{III}$-hs IS values at the B sites exhibit a linear shift with temperature. A similar

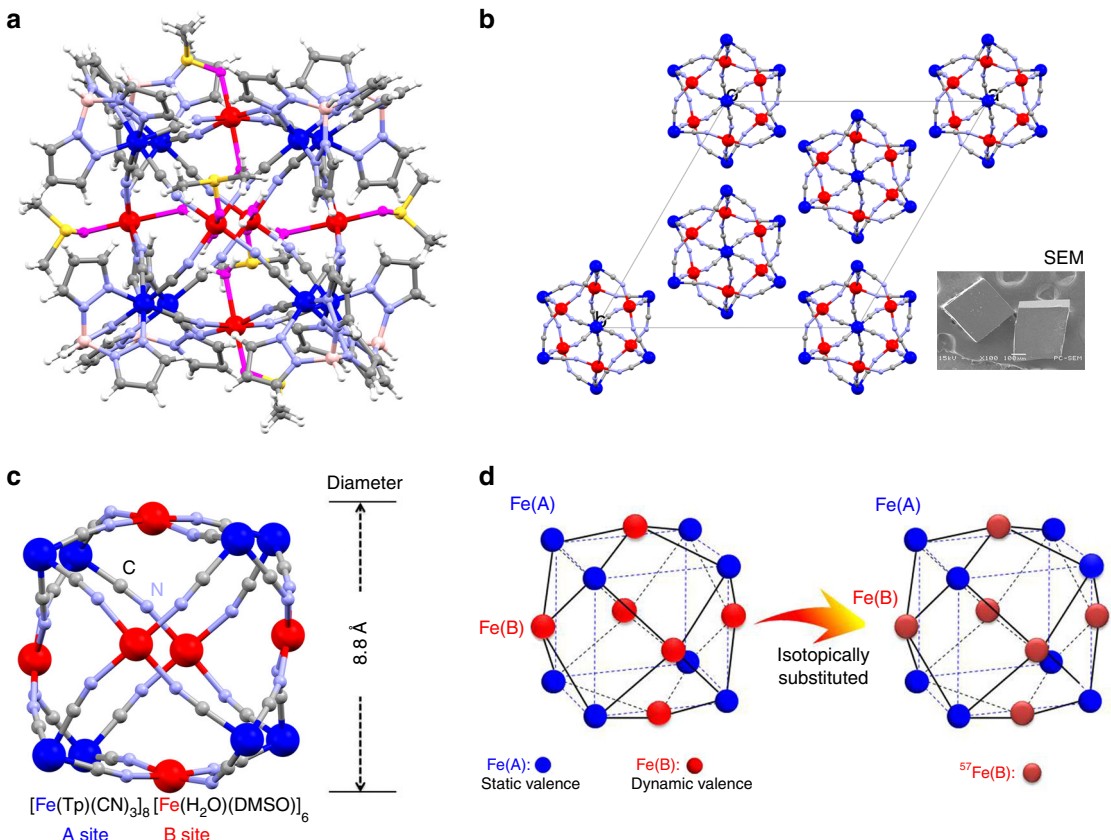

**Fig. 1** X-ray crystal structure determination. **a** Complete structure of the [Fe$_{14}$] spherical complex in a ball-and-stick style. The valence state on different Fe sites is indicated by two-color systems, that is, A-site Fe (blue) and B-site Fe (red), and O (pink), S (yellow), N (purple), C (gray), and H (white). **b** View of the cubic packing of the simplified [Fe$_{14}$] complex along the crystallographic c-axis. The inset denotes the scanning electron microscope (SEM) image of a single crystal of [Fe$_{14}$], illustrating the cubic faces. **c** The simplified [Fe$_{14}$] atomic layer framework with a diameter of 8.8 Å. **d** B-site-selective isotope substitution in the molecule.

behavior has previously been reported in a mixed-valence trinuclear Fe complex, [Fe$^{II}$Fe$^{III}$$_2$O(CH$_3$COO)(H$_2$O)$_3$][27], where the slope of the second-order Doppler shift was calculated as $6.3 \times 10^{-4}$ mm s$^{-1}$ T$^{-1}$, comparable to the value of $5.6(3) \times 10^{-4}$ mm s$^{-1}$ K$^{-1}$ for [Fe$_{14}$]. The electron-hopping relaxation model provides a good fit in the temperature range of 164–297 K, with a relaxation time of $\tau$ as shown in Fig. 2e. A typical Mössbauer time window was observed to range from ~$3 \times 10^{-7}$ to ~$1 \times 10^{-8}$ s for the system. Actually, in a previous study using the relaxation model, it was indicated that such dynamics are detectable as spectral changes in the range of ~$6.5 < -\log(\tau/s) < ~8.0$[28], which is comparable to the case of [Fe$_{14}$]. The electron-hopping activation energy ($U_{\text{eff}}$) was successfully evaluated using the Arrhenius equation, $\ln(\tau) = \ln(\tau_0) + U_{\text{eff}}/k_BT$. The best fit of the experimental data yielded $U_{\text{eff}} = 943(47)$ cm$^{-1}$ with a pre-exponential factor of $\tau_0 = 5.0(16) \times 10^{-11}$ s. Hence, the temperature dependence of the rates of intramolecular electron transfer in the Mössbauer spectroscopic analysis clearly demonstrated that six B-site Fe ions form a class II mixed-valence system according to the Robin–Day classification[29]. Electronic spectroscopy in the domain of intervalence transitions exhibits a wide peak centered at 11,100 cm$^{-1}$ (900 nm) with a pronounced absorption tail due to the superposition of adjacent IVCT (Fe$^{II}$-CN-Fe$^{III}$) and remote IVCT (Fe$^{II}$-NC-Fe$^{II}$-CN-Fe$^{III}$), as indicated in the reported cyanide-bridged systems with N-terminal mixed-valence state[24,30,31]. (Supplementary Fig. 3)

The crystal structure was additionally analyzed to probe whether a temperature-induced charge separation occurs in [Fe$_{14}$] at lower temperature, that is, 25 K, a temperature at which

the electrons should be completely frozen ($\tau \sim 10^{13}$ s) according to aforementioned Arrhenius equation. However, no crystallographic differences in the coordination environment of the B-site Fe ions were observed between the structures at low and room temperature. At 25 K, the B-site Fe ions stay at the crystallographically identical sites in the unit cell as in the case of the RT (room temperature) structure. Therefore, although two Fe$^{II}$ are expected to be located in the *trans* positions of the [Fe$_6$] octahedron to minimize Coulombic repulsion, the positions of the Fe$^{II}$ atoms cannot be assigned by X-ray diffraction. This is mostly due to the random distribution of the localized electrons at six B-site irons in the lattice. Thus, the charges at each B-site Fe virtually hold site occupation factors of 0.333 for Fe$^{II}$ and 0.667 for Fe$^{III}$, respectively. The heat capacity of [Fe$_{14}$] under zero field at 7–300 K was investigated (Fig. 3a). No distinct anomaly was observed in the heat capacity data, further indicating no first- or second-order phase transition. The observation is consistent with the thermal dependence of the rate of intramolecular electron transfer as indicated in the Mössbauer spectra.

To elucidate the electron-hopping behavior in the [Fe$_{14}$] complex, temperature-dependent infrared (IR) spectroscopic measurements were carried out at 78–350 K. The IR spectrum of [Fe$_{14}$] at 78 K exhibits several strong $\nu_{\text{CN}}$ stretches at 2101, 2078, 2068, 2053, and 2042 cm$^{-1}$ (Fig. 3b), indicative of the Fe$^{II}$-ls species bound to the C terminal of a cyanide bridge and both Fe$^{II}$ and Fe$^{III}$ to the terminal N atom. Notably, only two $\nu_{\text{CN}}$ stretching bands were observed in the same region for a previously reported [Fe$_{42}$] cluster, [{Fe(Tp)(CN)$_3$}$_{24}${Fe(H$_2$O)$_2$}$_6${Fe(dpp)

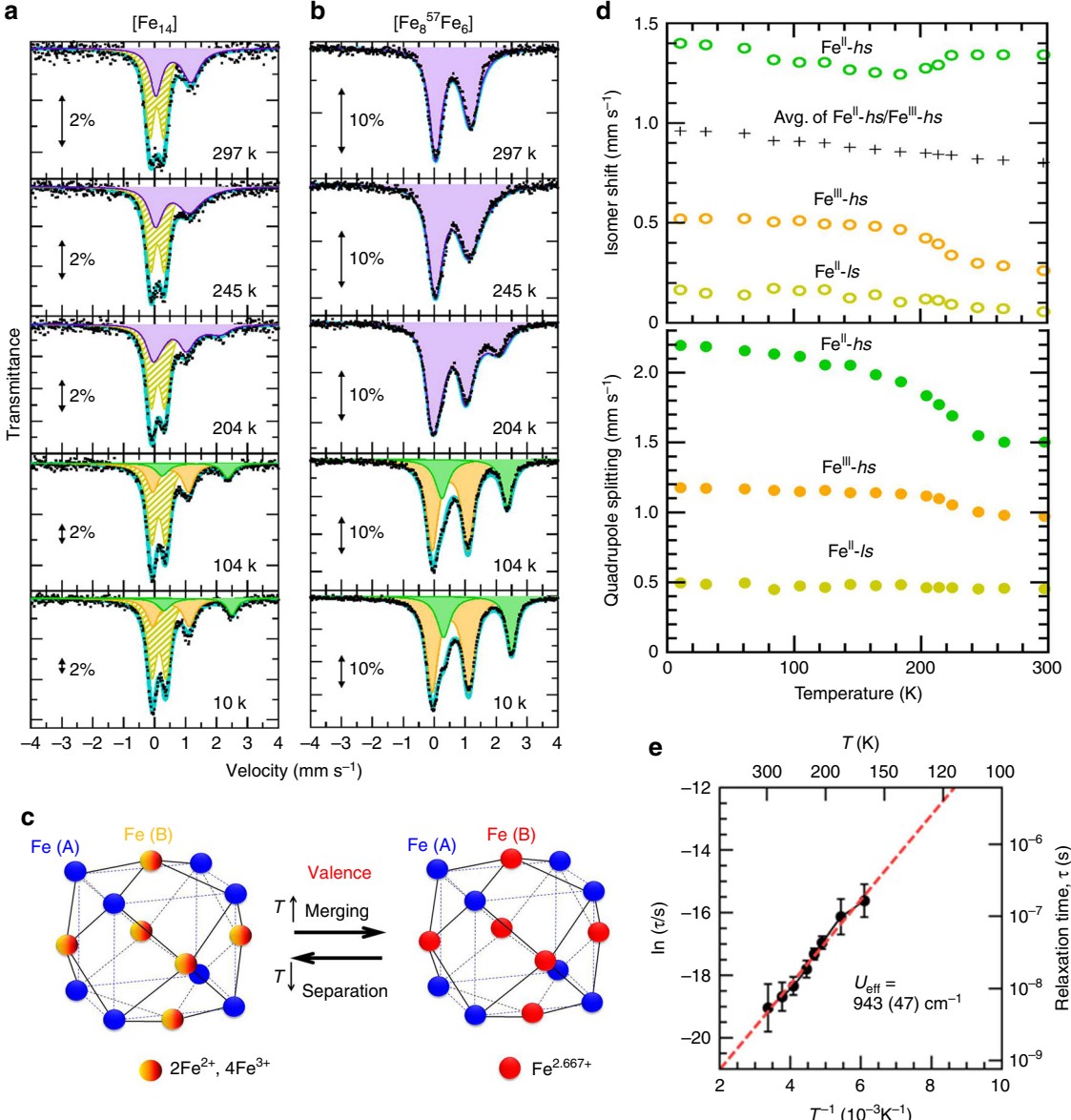

**Fig. 2** Mössbauer spectra analysis of the valence and spin state of the Fe ions. **a–b** Zero-field $^{57}$Fe Mössbauer spectra of a natural isotopic [Fe$_{14}$] and a $^{57}$Fe-enriched sample on the N-terminal B site, [Fe$_8$$^{57}$Fe$_6$], at selected temperatures with fits considering the electron hopping between Fe$^{II}$-hs and Fe$^{III}$-hs (purple area), Fe$^{II}$-hs (green area), Fe$^{III}$-hs (orange area), and Fe$^{II}$-ls (striped yellow area). **c** Schematic of the valence behavior on the crystallographically equivalent B site in the Mössbauer time window. **d** Quadrupole splitting (QS) and isomer shift (IS) as a function of the temperature in the Mössbauer spectra. The values of Fe$^{II}$-hs (green) and Fe$^{III}$-hs (orange) are determined from the $^{57}$Fe-enriched sample, whereas those of Fe$^{II}$-ls (striped yellow area) are determined from the natural sample. **e** Temperature dependence of the relaxation time ($\tau$) of the electron hopping between Fe$^{III}$-hs and Fe$^{II}$-hs in the temperature window of 164–297 K. The error bars were estimated as the respective deviations having the deference of the reduced chi-square value ($\Delta\chi^2$) within 2, considering the parameter correlations in the fitting. The red line represents the Arrhenius fitting in the high-temperature region.

(H$_2$O)}$_{12}$(CF$_3$SO$_3$)$_6$]·18H$_2$O    (dpp = 1,3-di(4-pyridyl)propane), wherein the valence-trapped Fe$^{II}$-ls–CN–Fe$^{III}$-hs structure was preserved across the whole temperature range[32] (Supplementary Fig. 4). Furthermore, the $\nu_{CN}$ stretching frequency in Prussian blue with a Fe$^{II}$-ls–CN–Fe$^{III}$-hs structure has been previously observed to be slightly higher compared with that in Prussian white with a Fe$^{II}$-ls–CN–Fe$^{II}$-hs structure[33]. The additional $\nu_{CN}$ stretching bands for [Fe$_{14}$], which lack in [Fe$_{42}$], indicates the presence of a Fe$^{II}$-ls–CN–Fe$^{III}$-hs and Fe$^{II}$-ls–CN–Fe$^{II}$-hs mixed linkage, by in situ Fourier-transform infrared spectroscopy. These results mean that the electron-hopping rate is slower than the time scale of the IR technique (10$^{-12}$–10$^{-13}$ s)[34].

Iron L-edge X-ray absorption spectra (XAS) were measured between 3.5 and 300 K to further characterize the electronic structure (Fig. 3c). Spin–orbit coupling of the 2$p^5$ final state (2$p^6$ 3$d^n \rightarrow$ 2$p^5$ 3$d^{n+1}$) leads to the splitting of L$_{2,3}$ edges into L$_3$ ($J = 3/2$) and L$_2$ ($J = 1/2$) absorption regions, which are separated in energy by ~12 eV. The L$_3$ and L$_2$ absorption edges of [Fe$_{14}$] were found to have maxima at 708.5 and 722 eV, respectively. The [Fe$_{14}$] L edge includes overlapping contributions from the A and B sites. The XAS spectra measured between 3.5 and 300 K are identical, confirming that there is no resolvable change in the local Fe coordination symmetry or crystal field splitting at the A or B sites between the low- and high-temperature states.

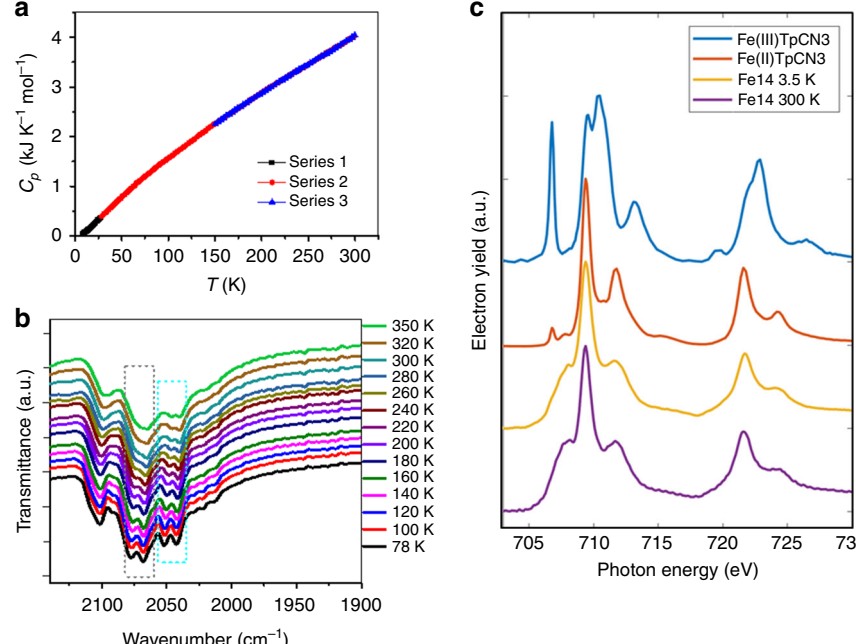

**Fig. 3** Thermal dependence of the intramolecular electron transfer rate. **a** Zero-field molar heat capacity of the $[Fe_{14}]$ sample at temperatures of 7–300 K. The series 1, 2 and 3 denote the repeatability of the data in three-testing in the different temperature ranges. **b** In situ temperature-dependent infrared spectra of $[Fe_{14}]$ in the solid state. The dashed rectangles are guided for view to include the changing bands with temperature. **c** Stacked L-edge XAS for the $[Fe_{14}]$ complex measured at 300 and 3.5 K alongside the reference spectra of $[Fe^{III}(Tp)(CN)_3]$ and $[Fe^{II}(Tp)(CN)_3]$. The $[Fe^{II}(Tp)(CN)_3]$ overlaps with all features in **c** within the $[Fe_{14}]$ complex, whereas the $[Fe^{III}(Tp)(CN)_3]$ does not (in particular the peak at ~706.8 eV associated with transitions into the $t_{2g}$ hole).

Furthermore, no temperature-induced changes in the spectra were observed for the $K\beta_{1,3}$ emission spectroscopy and high-resolution K-edge X-ray absorption near-edge structure measurements (Supplementary Fig. 5). The lack of evidence for the thermal transition by these techniques indicates that the electron-hopping rate is slower than the time scale of the X-ray spectroscopy measurements ($10^{-14}$ – $10^{-15}$ s). A comparison of the $[Fe_{14}]$ L-edge XAS data with the monomeric references $[Fe^{III}(Tp)(CN)_3]$ and $[Fe^{II}(Tp)(CN)_3]$ indicates that the valences of the eight A sites are diamagnetic $Fe^{II}$-ls[35–38], confirming the Mössbauer assignment.

**Magnetic properties**. To assess the magnetic properties of $[Fe_{14}]$, variable-temperature direct current magnetic susceptibility data were collected. Figure 4a shows that at 300 K the $\chi_M T$ value is 27.45 cm$^3$ K mol$^{-1}$. Upon a decrease in the temperature, the $\chi_M T$ value shows a steady increase, before beginning to abruptly increase at ca. 30 K with a maximum of 59.17 cm$^3$ K mol$^{-1}$ at 3.5 K. This magnetic behavior suggests an overall ferromagnetic intramolecular exchange for $[Fe_{14}]$. The data in the range of 300–10 K can be fitted according to the Curie–Weiss law, yielding $C = 27.17$ cm$^3$ K mol$^{-1}$ and $\theta = 2.95$ K. This $C$ value is comparable to the contribution from two $S = 2$ $Fe^{II}$-hs, four $S = 5/2$ $Fe^{III}$-hs, and eight $S = 0$ $Fe^{II}$-ls centers in total for an individual $[Fe_{14}]$ complex, which is further supported by the evaluated experimental magnetic entropy 87.1 J K$^{-1}$ mol$^{-1}$ $\approx R\ln(5^2 \times 6^4)$ (=86.4 J K$^{-1}$ mol$^{-1}$, expected magnetic entropy). The positive Weiss constant suggests an intramolecular ferromagnetic interaction[39]. Density functional theoretical (DFT) calculations revealed that the spin density is mainly concentrated on the six B-site Fe ions (avg. spin value of 3.535) rather than the A-site Fe ions (avg. spin value of 0.232) (Supplementary Fig. 6), which is consistent with the experimental observations. To determine the spin ground state of $[Fe_{14}]$, magnetization data were collected and were plotted as reduced magnetization

$(M/N\mu_B)$ vs. applied field in the range of 0–5 T at 2 K, as shown in Fig. 4b. The onset of magnetization in the applied field is greater than that of six magnetically isolated centers (red line in Fig. 4b) and is closer to the Brillouin curve for one isotropic $S = 14$ center (blue line in Fig. 4b). Inclusion of weak ferromagnetic exchange coupling (green line) between B sites suitably reproduces the measured magnetization curve, indicating a ferromagnetic $S = 14$ ground state.

The presence of a ferromagnetic interaction was also confirmed using X-ray magnetic circular dichroism (XMCD), a technique that is a probe of magnetization at the atomic level. Since they are diamagnetic, the A-site Fe ions have no XMCD intensity. Hence, the $[Fe_{14}]$ XMCD spectrum provides direct spectroscopic access to only the B-site Fe ions. The left and right circularly polarized total fluorescence yield (TFY) detected XAS at 3.5 K, and 14 T is shown in Fig. 4c, alongside the XMCD spectra at various applied magnetic fields. At 3.5 K, the B sites are in the valence-trapped state in a ratio of 4:2 ($Fe^{III}$ vs. $Fe^{II}$). The onset of the $L_3$ edge for the octahedral $Fe^{II}$-hs component is known to be lower in energy than that of $Fe^{III}$-hs by ~1 to 2 eV. Hence, the lower energy region of the $L_3$ XMCD spectrum is expected to exhibit $Fe^{II}$-hs contributions with respect to the higher energy region, which is expected to show $Fe^{III}$-hs contributions. The magnitude of the XMCD signal across the $L_3$ edge increases monotonically with an applied field indicating that the ferrous and ferric spins at the B sites are ferromagnetically exchange coupled[40]. (Supplementary Fig. 7) The field dependence of the XMCD data is in accordance with that observed for the superconducting quantum interference device (SQUID) magnetization.

It should be noted that the maximum $\chi_M T$ at low temperatures is significantly reduced compared to the expected $\chi_M T$ value for an isolated $S = 14$ ground state (~105 cm$^3$ K mol$^{-1}$). (Supplementary Fig. 8) The susceptibility measurements conducted down to 0.5 K exhibits a sharp peak in $\chi_M$ at 0.8 K (Fig. 4d), showing a

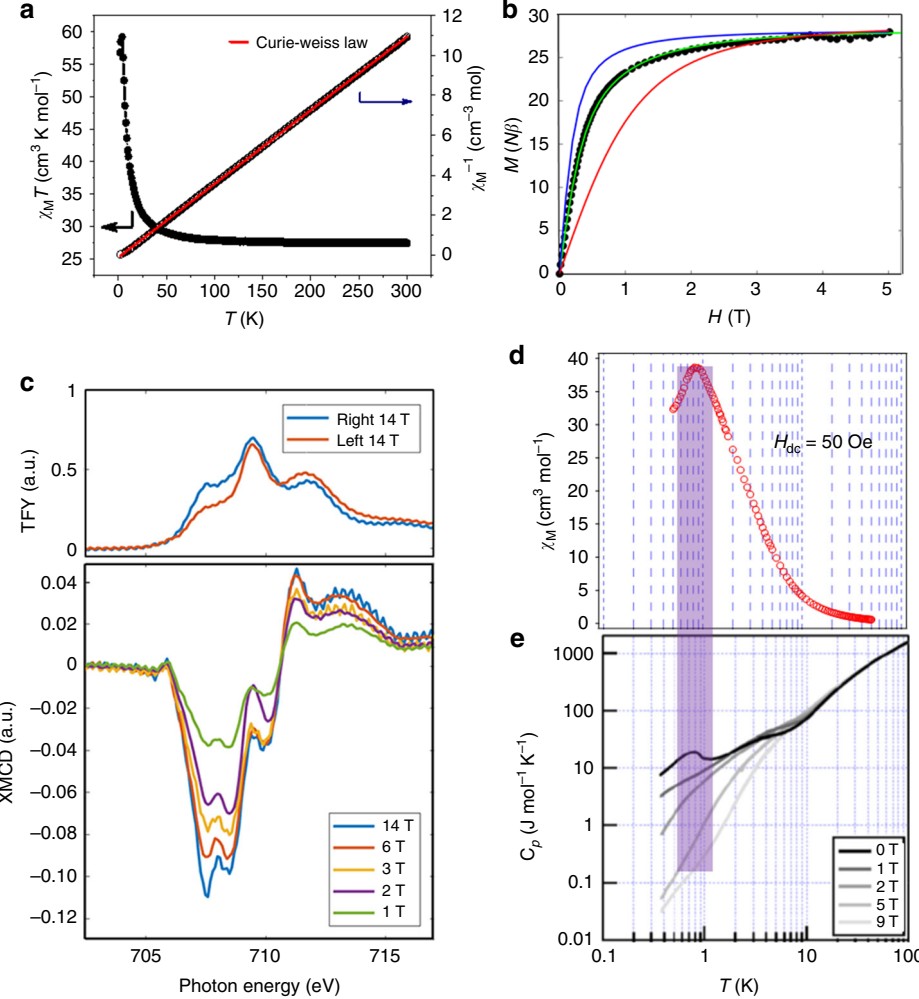

**Fig. 4 Magnetic characterization. a** Temperature dependence of $\chi_M T$ and $\chi_M^{-1}$ for [Fe$_{14}$] ($H_{dc}$ = 100 Oe). The black symbol is the experimental data and the red line is the fitting result. **b** Magnetization vs. external magnetic field curve for [Fe$_{14}$] at 2 K. The blue line corresponds to the calculated Brillouin function for $S = 14$ ($J \rightarrow +\infty$). The red line represents the sum of the Brillouin functions that correspond to four magnetically isolated $S = 5/2$ and two $S = 2$ ($J = 0$). The green line represents the simulated curve that includes weak ferromagnetic coupling between B sites ($J = 0.9$ K, $g = 2.0$). **c** Top: the left and right circularly polarized L$_3$-edge total fluorescence yield spectra of the [Fe$_{14}$] complex (14 T, 3.5 K). Bottom: the field dependence of the [Fe$_{14}$] L$_3$-edge XMCD spectra (left–right) measured at 3.5 K. **d** Magnetic susceptibility under a 50 Oe field in the temperature region of 50–0.5 K. **e** The field-dependent heat capacity of [Fe$_{14}$] from 0 to 9 T in the temperature region of 100–0.36 K.

typical feature of antiferromagnetic long-range magnetic ordering. Heat capacity measurements were carried out and showed a λ-type anomaly centered at $T_N = 0.85$ K in the zero field (Fig. 4e), revealing the onset of weak long-range magnetic ordering. The magnetic nature of this feature was confirmed by its disappearance upon the application of a strong magnetic field. The intermolecular interaction was also suggested by the high-frequency electron paramagnetic resonance spectra of [Fe$_{14}$] (Supplementary Fig. 9). Since the high-spin sites are coordinated via diamagnetic [Fe$^{II-ls}$(Tp)(CN)$_3$], the mechanism of intramolecular ferromagnetic exchange over the atomic layer of [Fe$_{14}$] necessitates the delocalization of the $t_{2g}$ [Fe$^{II-ls}$(Tp)(CN)$_3$] metal character to B-site $t_{2g}$ orbitals via bridging CN π* orbitals. A theoretical description of the intramolecular exchange on the atomic layer of [Fe$_{14}$] requires consideration of the electron delocalization, as described by Mayoh and Day[41,42]. In practice, this requires the evaluation of electron transfer integrals and interelectronic Coulombic repulsions, in conjunction with the Heisenberg contribution to the exchange for charge transfer states[43]. However, the value of such an analysis is limited due to the onset of 3D magnetic ordering at low temperature.

In summary, we report on the synthesis and electronic behavior of a nanosize [Fe$_{14}$] complex with the two-electron, four-hole mixed-valence state. Two extra 3$d$ electrons in the complex are found to hop at N-terminal Fe centers on its atomically thin spheric surface. The valence of the A-site Fe is basically static, while the valence electrons on the B-site Fe exhibit thermal dependence of the rate of intramolecular electron transfer. Furthermore, [Fe$_{14}$] has a high-spin ground state with $S = 14$, meaning that the two electrons are hopping around the exchange-coupled atom-layer thin surface. The nanoarchitecture in this work may be useful for application in future molecular electronic and chemical devices using the intermetallic electronic and magnetic interactions in the framework and precise nanospace.

## Methods

**Synthesis of the [Fe$_{14}$] complex.** A 2 mL aqueous solution containing 6 μmol of Fe(BF$_4$)$_2$·6H$_2$O was layered over a 2 mL DMSO aqueous solution containing 8 μmol of Bu$_4$N[Fe(Tp)(CN)$_3$]. A 1 μmol aqueous solution of L-ascorbic acid was utilized as the middle buffer layer under aerobic conditions. Crystals suitable for X-ray diffraction analysis were obtained in a yield of 16% after a week. Anal. calcd. for

$C_{108}H_{128}B_8Fe_{14}N_{72}O_{12}S_6$: C, 35.18; H, 3.50; N, 27.35. Found: C, 35.12; H, 3.52; N, 27.40. The analysis of Fe content in the solid sample of $[Fe_{14}]$ was performed through inductively coupled plasma atomic emission spectroscopy. The experiment result is 21.27 wt.%, which is in good agreement with the calculated value of 21.20 wt.%.

The $^{57}$Fe isotopically enriched sample, $[Fe_8{}^{57}Fe_6]$, was prepared in a similar manner to that of $[Fe_{14}]$, except that the isotopically enriched salt, $^{57}FeCl_2\cdot4H_2O$ ($^{57}$Fe, 96%), was utilized instead of $Fe(BF_4)_2\cdot6H_2O$. Through X-ray diffraction measurements, the crystal structure of the isotopically enriched sample, $[Fe_8{}^{57}Fe_6]$, was observed to be identical to that of the natural isotopic $[Fe_{14}]$.

**X-ray crystal structure determination**. X-ray diffraction data at room temperature and 123 K for $[Fe_{14}]$ were collected on a BRUKER APEX-II CCD (Bruker Corp.) equipped with a graphite-monochromated Mo-Kα radiation source ($\lambda = 0.71073$ Å)[44–47]. Diffraction data at 25 K were collected under a cold helium gas stream on a Rigaku HPC X-ray diffractometer, using multi-layer mirror monochromated Mo-Kα radiation ($\lambda = 0.71073$ Å). Bragg spots were integrated using the CrysAlisPro program package, and empirical absorption correction (multi-scan) was applied using the SCALE3 ABSPACK program. The structures were solved by direct methods (SHELXT Version 2014/4) using full-matrix least-squares refinement (SHELXL Version 2018/1)[48]. The H atoms were geometrically placed on organic ligands in riding mode, and all of the non-H atoms were anisotropically refined by full-matrix least-squares refinement on $F^2$ using the SHELXTL program[49]. A summary of the crystallographic data and refinement parameters is presented in Extended Data Table 1.

**Computational details**. Spin-polarized DFT calculations were carried out using the PWscf module in the Quantum Espresso 6.1 program package[50]. The exchange and correlation term in the Kohn−Sham equation was approximately treated using the Perdew–Burke–Ernzerhof method in terms of the gradient of the electronic density[51]. Projector-augmented wave potentials[52] were used along with a plane-wave basis set with a kinetic energy cut-off of 31.0 Ry. The simulation cell of $[Fe(Tp)(CN)_3]_8[Fe(H_2O)(DMSO)]_6$ contained 348 atoms in total. Atomic optimizations were carried out, starting from the experimental cell parameters. Integration in the first Brillouin zone for geometry optimizations was performed using $2 \times 2 \times 2$ point Monkhorst–Pack sampling[53]. The SCF (self-consistent field) convergence and the total force convergence were set to be $1.0 \times 10^{-6}$ (Ry) and $1.0 \times 10^{-3}$ (Ry/au), respectively. The total magnetization was constrained to be 28.00 Bohr mag per cell. The spin-polarized DFT calculations further provided reliable information as to the charge and the spin state on each iron atom in $[Fe_{14}]$. The singly occupied electrons at the B-site $Fe^{III}$ components partially delocalizes on the A-site $Fe^{II}$ ions.

**Magnetic analysis**. The magnetic measurements of the samples were performed using a SQUID (MPMS-5S) magnetometer (Quantum Design Inc., USA). The magnetic susceptibility measurements shown in Fig. 4d were performed using MPMS-XL7AC (Quantum Design Inc., USA) apparatus with a 9 mm diameter $^3$He insert[54]. The data were corrected for diamagnetic contributions, calculated using Pascal's constants[55].

**Simulation of magnetization**. The field dependence of magnetization for $[Fe_{14}]$ follows neither the Brillouin function for an $S = 14$ spin moment or the sum of uncoupled moments (four $S = 5/2$ and two $S = 2$). Inclusion of a weak ferromagnetic exchange interaction, acting between high spin sites was found to reproduce the measured magnetization curve. The magnetization curve was fit assuming one exchange constant $J = 0.9$ K and $g = 2.0$, based on the following Hamiltonian:

$$\widehat{H} = -2J(S_1S_2 + S_1S_3 + S_1S_4 + S_1S_5 + S_2S_3 + S_2S_5 + S_2S_6 + S_3S_4 \\ + S_3S_6 + S_4S_5 + S_4S_6 + S_5S_6) + \sum_i \mu_B\mathbf{B}\bar{g}_iS_i,$$

where $S_1$ and $S_6$ represent the $S = 2$ sites and $S_2$, $S_3$, $S_4$ and $S_5$ represent the $S = 5/2$ sites, $B$ is the applied magnetic field and $\mu_B$ is the Bohr magneton. This exchange model neglects the double-exchange component to the Hamiltonian present in Class II systems and effects due to dipolar fields from neighboring molecules. However, this simplified model indicates how the measured curve is consistent with weak ferromagnetic exchange coupling within $[Fe_{14}]$.

**$^{57}$Fe Mössbauer spectroscopy**. The $^{57}$Fe Mössbauer spectra were measured using a conventional Mössbauer spectrometer (Topologic Systems, Kanagawa, Japan) in transmission mode with a $^{57}$Co/Rh γ-ray source. Low-temperature measurements were performed upon a CryoMini/CryoStat cryogenic refrigerator set (Iwatani Industrial Gases, Osaka, Japan). The samples were tightly sealed with silicon grease in an acrylic holder and the spectra were calibrated using α-Fe foil as a reference at room temperature. The spectral fitting was carried out using the MossWinn 4.0 program and the full zero-field $^{57}$Fe Mössbauer spectra at all investigated temperatures were provided in Supplementary Fig. 10. For the $^{57}$Fe-enriched $[Fe_8{}^{57}Fe_6]$ sample, the spectra were analyzed by applying an electron-hopping relaxation model (164–297 K; see Supplementary Method and Supplementary

Fig. 11) or two quadrupole doublets (10–144 K). The area ratio was fixed at the ideal value ($Fe^{II}/Fe^{III} = 1/2$) according to the chemical formula in order to avoid overparameterization. Some parameter correlations were found, especially between the linewidth and relaxation rate, through curve fittings of the spectra at 245–297 K. Therefore, the linewidth and IS values for $Fe^{II}$-hs at the relevant temperatures were fixed using those at 224 K. For the analyses at 10–144 K, the $Fe^{II}$-hs doublet was regarded as an asymmetric doublet, rising as a result of paramagnetic relaxation because an alternative symmetric doublet was tried but presented no sufficient result. For the natural isotopic sample of $[Fe_{14}]$, the Mössbauer spectra were analyzed using an additional doublet ($Fe^{II}$-ls; site A) together with the corresponding $Fe^{II}$-hs and $Fe^{III}$-hs signals (site B) reproduced by the fixed parameters obtained from $[Fe_8{}^{57}Fe_6]$. Across the entire temperature range, the $Fe^{II}$-ls/$Fe^{III}$-hs/$Fe^{II}$-hs ratio of the sample was maintained at 8/4/2.

**Heat capacity calorimetry**. Heat capacity measurements were performed with a laboratory-made adiabatic microcalorimeter in the temperature range of 9–300 K (adiabatic method) and with a PPMS (Quantum Design Inc., USA) in the temperature range 0.36–100 K under magnetic fields of 0–9 T (relaxation method). In the adiabatic calorimetry, 0.06356 g of a polycrystalline sample, which was made a buoyancy correction, was loaded into a 0.09 cm$^3$ gold-plated copper cell and sealed with an indium wire under helium gas atmosphere. Thermometry was performed using a rhodium–iron alloy resistance thermometer (nominal 27Ω, Oxford Instruments) calibrated on the basis of the international temperature scale of 1990 (ITS-90). In the relaxation calorimetry (PPMS), we used buoyancy-corrected 1.1031 mg of a polycrystalline sample formed into a pellet of 2.5 mm in diameter. For the measurements below 10 K, a $^3$He insert was employed[56].

**X-ray absorption spectra and X-ray magnetic circular dichroism**. XAS and XMCD measurements at the Fe L absorption edges (703–740 eV) were measured on beamline I10 at the synchrotron Diamond Light Source of the Harwell Science and Innovation Campus in Oxfordshire in the United Kingdom. The XMCD spectra were obtained by flipping the helicity of circularly polarized X-rays exhibiting a 100% degree of polarization in the case of fixed applied magnetic fields. The measurements were performed with the temperature of the sample holder being regulated between 3.5 and 300 K. The total electron yield was obtained by measuring the drain current of the sample, whereas the TFY was obtained using a photodiode. The powdered samples of $[Fe_{14}]$ were attached with indium, to a copper sample holder. Radiolysis was controlled through the attenuation of the incident X-ray flux to 7% of the optimized value. Multiple scans were performed at each sample location to maintain control of the radiolysis, which was indicated by an increase in intensity at the low-energy portion of the $L_3$ edge. TFY-detected measurements were found to be less susceptible to radiolysis and were hence adopted for XMCD measurements.

**High-frequency electron paramagnetic resonance**. High-frequency electron paramagnetic resonance (HF-EPR) measurements were performed on a locally developed spectrometer at the Wuhan National High-magnetic Field Center with a pulsed magnetic field of up to 30 T.

## Data availability

The data that support the findings of this study and its Supplementary Information are available from the corresponding author (D.W. or O.S.) upon reasonable request. The X-ray crystallographic coordinates for structures reported in this paper have been deposited at the Cambridge Crystallographic Data Center (CCDC), under deposition number CCDC 1,878,752–1,878,754. These data can be obtained free of charge from The Cambridge Crystallographic Data Center via www.ccdc.cam.ac.uk/data_request/cif.

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

## Acknowledgements

We are thankful for the financial support by the PAPD of Jiangsu Higher Education Institutions. This work is supported by the NSFC programs (Grants 21671027, 21471023 and 21671172), by MEXT KAKENHI (Grant Number JP17H01197, JP17H03117, JP16K05725 and JP18K05057), JST-CREST "Innovative Catalysts" JPMJCR15P5) and sponsored by the Jiangsu Provincial QingLan Projects. D.W. thanks Prof. Z.W. Ouyang and Dr. Z.X. Wang for assisting the HF-EPR measurements with the support by Wuhan National High Magnetic Field Center. XAS and XMCD experiments were carried out with the support of the Diamond Light Source (proposal SI17723). A portion of this work was performed on the Steady High Magnetic Field Facilities, High Magnetic Field Laboratory, CAS.

## Author contributions

D.W. and O.S. conceived the experiment and wrote most of the paper. W.H., S.W. and S.K. prepared the samples. X.G. and Y.L. performed IR and UV–visible spectra measurement at low temperature. M.L.B. and P.B. measured the XAS and XMCD spectra and

discussed the results. A.O., S.H. and N.K. undertook the spectra of Mössbauer and fitted the data. M. Noguchi, Y.M. and M. Nakano performed heat capacity measurement. T.N. measured single-crystal structure at low temperature. Y.I. and T.K. performed magnetic experiment at low temperature. G.-L.Z., Y.S. and K.Y. conducted DFT calculation. All the authors discussed the results and commented on the paper.

## Competing interests

The authors declare no competing interests.
